# Assessment of Energy Consumption of Brine Discharge from SWRO Plants

**Rubén Navarro [1], José L. Sánchez Lizaso [2,\*] and Iván Sola [2,3]**

[1] Mancomunidad de los Canales del Taibilla, Ministerio para la Transición Ecológica y el Reto Demográfico, 30008 Cartagena, Spain

[2] Department of Marine Sciences and Applied Biology, University of Alicante, San Vicente del Raspeig s/n, 03690 Alicante, Spain

[3] HUB-AMBIENTAL UPLA, Universidad de Playa Ancha, Valparaíso 2360072, Chile

[\*] Correspondence: jl.sanchez@ua.es

**Abstract:** The San Pedro del Pinatar I and II desalination plants in Spain were constructed near *Posidonia oceanica* meadows protected at the national and European level. The environmental impact statement for these plants stipulate that the brine discharge from the plant must not impact the meadows. To this end, a 4790 m submerged outfall was constructed to bypass the lower limit of the seagrass meadows, and a diffuser piece, along with an outfall pumping system, was installed at the end of the outfall. The objective of this paper is to evaluate the economic cost of the energy consumed for the brine discharge evacuation process necessary to comply with environmental requirements. The operating time and power consumption data were obtained from the plant's monitoring system, while the energy cost was obtained from energy invoices. The computed results show that it is possible to minimize the environmental impacts of brine discharge on the marine environment of an SWRO plant with a low economic cost. The average energy consumption of the reject effluent pumping system ranged from 19.4 to 1239.3 thousand kWh per year, while the average annual energy cost was 49,329 €, which amounts to only 0.56% of the total energy cost for plant operation. The adoption of these measures provide a cost-effective means to meet environmental protection requirements and minimize the environmental impact associated with the discharged brine. As the demand for desalination operations increase, economically and scientifically viable technologies for mitigating environmental impacts are necessary for sustainability in this domain.

**Keywords:** brine discharge; submerged outfall; energy consumption; economic cost





## 1. Introduction

Seawater Reverse Osmosis (SWRO) is the most popular water treatment technology for desalination operations globally [1]. The SWRO process generates high-quality freshwater, but it also results in the production of a brine that should be discharged [2]. The environmental impact of brine discharges from SWRO plants on the marine environment is a key concern in desalination development [3,4].

Hypersaline effluents from SWRO plants are usually discharged into the sea, as this costs much less compared to other disposal methods [5]. Brine discharge can double the salinity of seawater intake and result in the formation of a high-density saline plume that tends to follow the slope of the seabed. Such high-salinity flows may harm benthic communities in the brine discharge area [6–10]. The dilution of brine discharge in a marine environment depends upon many factors such as: the disposal method (coastal or submarine outfall), hydrodynamic characteristics of discharge area (waves, currents and tides), and bathymetry of the seabed, among others [5,6,11,12]. Brine dilution in a marine environment can be increased by methods such as the use of diffusers (single or multiport diffusers), bypassing seawater before discharge, or mixing brine with other effluents [8,13–15]. However, the use of these measures can increase the energy consumption of desalination plants [16].

The San Pedro del Pinatar SWRO plants located on the southeast of the Spanish Mediterranean coast began operation in 2006 with a maximum installed capacity of 130,000 m$^3$, which represented a maximum brine discharge production of 159,000 m$^3$. They are located near *Posidonia oceanica* seagrass meadows that are protected by national and European regulations. The brine discharged from both plants is disposed into the sea through a 4790 m outfall to ensure that the discharge point is situated below the lower limit of the *Posidonia oceanica* meadow [17–19].

Initially, no further measures were considered in relation to this project for the protection of the marine environment. However, environment monitoring results showed high salinity values around the brine discharge area and negative effects on the abundance and diversity of benthic communities present there [8,20]. Therefore, in 2010, it was decided to install a diffuser piece at the end of the submerged outfall to maximize effluent dilution and mitigate the identified environmental impact [14]. Diffusers are devices used to maximize the dilution process of brine discharges with the nearby seawater, in order to enhance jet exit velocity and therefore the mixing process. The outfall of the San Pedro del Pinatar SWRO plants has only one diffuser at the end of the submerged outfall which creates an inclined dense jet to achieve maximum mixing process efficiency [14]. Subsequent monitoring indicated a drastic reduction in salinity values in the brine discharge area along with a significant recovery of the abundance and diversity of benthic organisms [8].

Even though the installation of diffuser pieces at the end of submerged outfalls reduces the environmental impact of desalination plants, it increases energy consumption as special equipment is required to increase the discharge pressure before the effluent can be released into the sea.

The objective of this study is to assess the energy consumption and economic cost associated with the installation of a diffuser piece at the end of the outfall of SWRO plants to meet environmental protection requirements and minimize the environmental impact associated with the discharged brine. Here the San Pedro del Pinatar SWRO plants are used as a case study.

## 2. Materials and Methods

### 2.1. Description of Brine Disposal

Brine discharged from both the desalination plants is collected in a single chamber (Figure 1). Three pumps, of which two have frequency variators and the third is fitted with a starter, are used to discharge the effluent into the chamber. The three pumps have a maximum flow rate of 4250 m$^3$/h and raise the brine discharge to a level of 7.5 m. Each pump has an electric motor with an installed power of 125 kW. The three pumps provide sufficient pressure to ensure that the brine is discharged through the diffuser piece to achieve effective dilution.

Once the discharged brine is pumped into the chamber, it is connected to the outfall, which has a diameter of 1400 mm at the submerged end. In order to improve the dilution of the brine, a diffuser piece with an angle of 60° and a final diameter of 700 mm (Figure 2) was installed in 2010. The discharged brine is disposed through the diffuser at 4.7 m depth from the seabed, which is at 35.7 m.

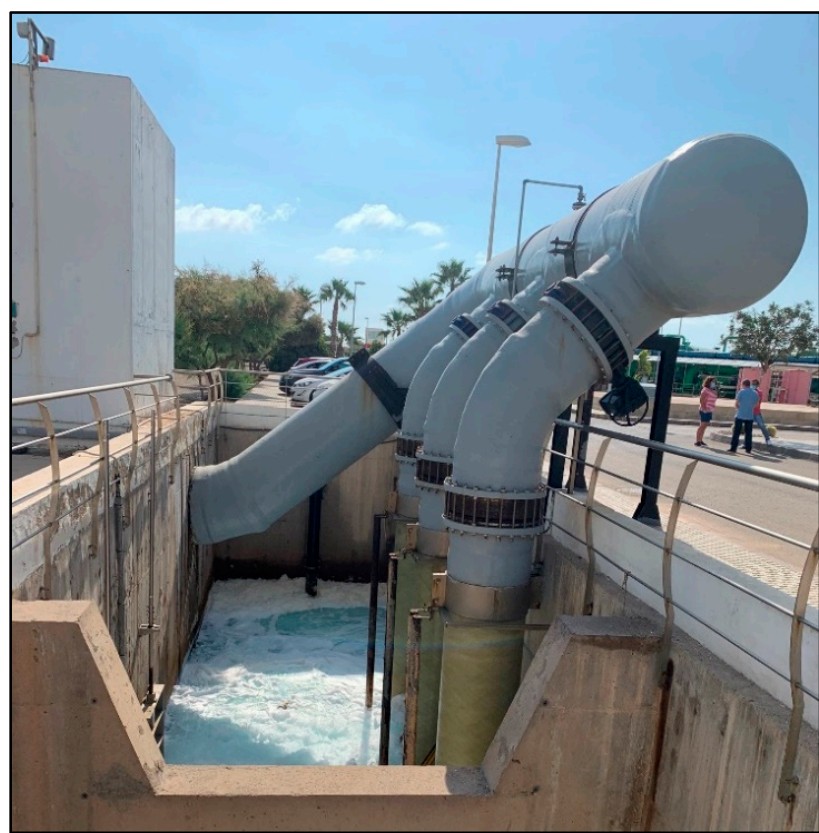

**Figure 1.** Brine discharge installation at San Pedro del Pinatar II SWRO plant. The effluent after the RO process is collected by the gray pipes driven by the pumps for disposal into the chamber where the outfall is located.

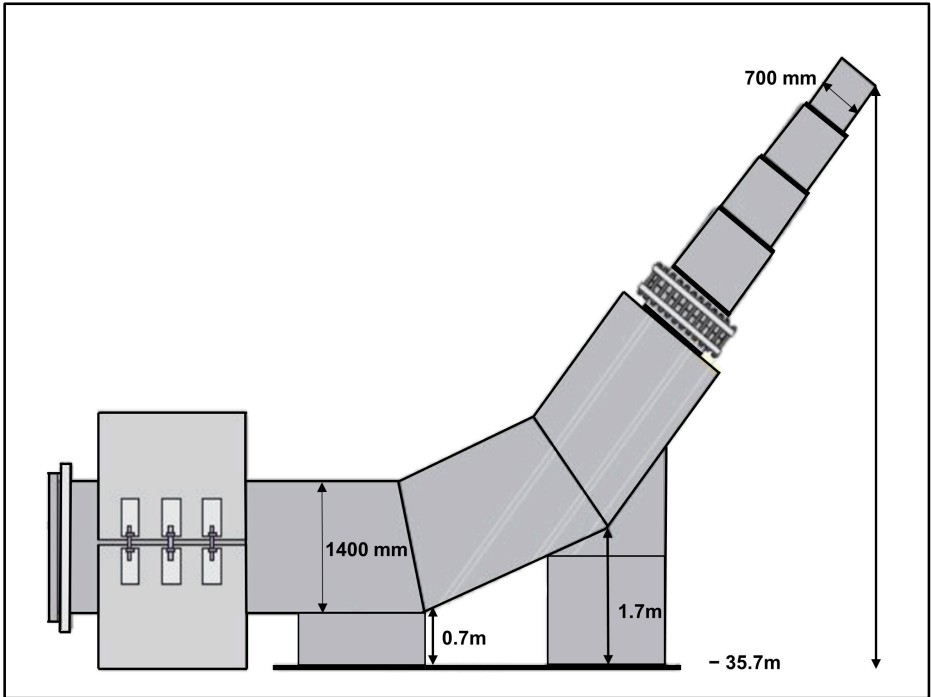

**Figure 2.** Schematic of diffuser device in the submerged outfall of San Pedro del Pinatar SWRO plants.

*2.2. Data Collection and Calculations*

The processes in the desalination plant are monitored and controlled using SCADA (Supervisory Control and Data Acquisition) software. SCADA records the operating time data of the three pumps used to evacuate the discharged brine. The power consumption data of the two pumps with frequency variators was recorded, while that of the third pump with the starter was calculated using a function directly proportional to the operating time.

The total economic cost of the pumping system was calculated based on energy invoices, and only the energy consumption was considered. The fraction of economic cost associated with the pumping system was calculated from the total energy cost and the ratio between the energy consumption of the pumps compared to the total energy consumption of both SWRO plants.

## 3. Results

*3.1. Freshwater Production*

Tables 1 and 2 summarize the monthly output of both desalination plants during the period under study (2012–2019). The months with the highest output of desalinated water were in the summer, between July and September, with an average production of 2.84 $Mm^3$.

**Table 1.** Freshwater production ($m^3$) of SWRO plant of San Pedro del Pinatar I.

| Month | 2012 | 2013 | 2014 | 2015 | 2016 | 2017 | 2018 | 2019 |
|---|---|---|---|---|---|---|---|---|
| January | 1,097,950 | - | 116,300 | 1,886,800 | 1,654,500 | 1,744,500 | 1,930,900 | 541,400 |
| February | 1,101,920 | - | 49,500 | 1,591,100 | 961,100 | 979,600 | 441,270 | 84,700 |
| March | 1,797,460 | 165,710 | 86,300 | 1,888,500 | 1,947,600 | 2,082,400 | 2,077,180 | 1,540,860 |
| April | 1,768,810 | 98,280 | 139,200 | 1,936,600 | 2,041,900 | 1,998,600 | 1,951,930 | 1,976,320 |
| May | 1,926,250 | 88,990 | 214,200 | 2,191,600 | 2,096,700 | 2,100,800 | 2,117,670 | 2,042,030 |
| June | 1,753,720 | 98,930 | 217,600 | 2,138,700 | 2,015,300 | 1,822,600 | 2,035,700 | 2,034,260 |
| July | 1,603,520 | 103,700 | 304,900 | 1,905,400 | 2,133,730 | 1,773,100 | 2,094,760 | 2,082,040 |
| August | 2,144,330 | 95,700 | 749,700 | 2,201,200 | 2,228,800 | 2,313,200 | 2,316,060 | 2,368,540 |
| September | 1,906,670 | 88,900 | 278,200 | 2,002,400 | 2,041,900 | 2,035,700 | 2,067,970 | 1,902,850 |
| October | 1,832,840 | 123,800 | 327,100 | 2,128,900 | 2,095,100 | 2,065,000 | 2,113,280 | 2,022,050 |
| November | 132,970 | 124,100 | 887,490 | 1,733,220 | 1,931,200 | 1,888,100 | 1,939,610 | 1,895,140 |
| December | 5600 | 116,800 | 1,673,000 | 1,603,680 | 1,772,100 | 1,923,500 | 1,869,250 | 1,735,450 |

**Table 2.** Freshwater production ($m^3$) of SWRO plant of San Pedro del Pinatar II.

| Month | 2012 | 2013 | 2014 | 2015 | 2016 | 2017 | 2018 | 2019 |
|---|---|---|---|---|---|---|---|---|
| January | 21,390 | 220,140 | 44,300 | 65,500 | 1,297,900 | 47,200 | 1,933,300 | 1,746,600 |
| February | 30,290 | 220,070 | 70,600 | 37,400 | 1,509,700 | 17,600 | 1,114,740 | 1,394,350 |
| March | 12,580 | 98,600 | 43,400 | 52,200 | 980,800 | 1,137,700 | 1,953,540 | 977,450 |
| April | 18,690 | 25,600 | 30,400 | 72,600 | 1,324,200 | 1,225,600 | 1,908,630 | 841,980 |
| May | 1,008,870 | 25,200 | 63,300 | 83,200 | 1,439,000 | 1,647,600 | 1,639,350 | 457,610 |
| June | 560,740 | 21,700 | 53,000 | 108,000 | 1,888,100 | 1,879,800 | 1,923,560 | 1,656,400 |
| July | 1,789,660 | 36,100 | 438,500 | 144,400 | 1,996,200 | 1,898,000 | 1,609,290 | 2,002,530 |
| August | 409,690 | 23,800 | 94,700 | 1,614,500 | 1,983,500 | 2,047,300 | 1,395,530 | 2,043,150 |
| September | 1,263,540 | 16,700 | 513,100 | 638,900 | 1,902,800 | 1,772,100 | 1,845,450 | 1,822,280 |
| October | 1,071,460 | 44,700 | 544,700 | 965,500 | 1,558,800 | 1,664,000 | 989,160 | 2,003,800 |
| November | 203,560 | 5400 | 240,800 | 571,700 | 920,600 | 1,896,100 | 668,860 | 1,934,120 |
| December | 184,670 | 13,700 | 207,800 | 672,600 | 239,600 | 2,008,900 | 798,220 | 1,285,940 |

Figure 3 presents the freshwater production of SWRO plants of San Pedro del Pinatar I and II. Since 2016, production has remained stable at an average of 39.76 $Mm^3$ per year.

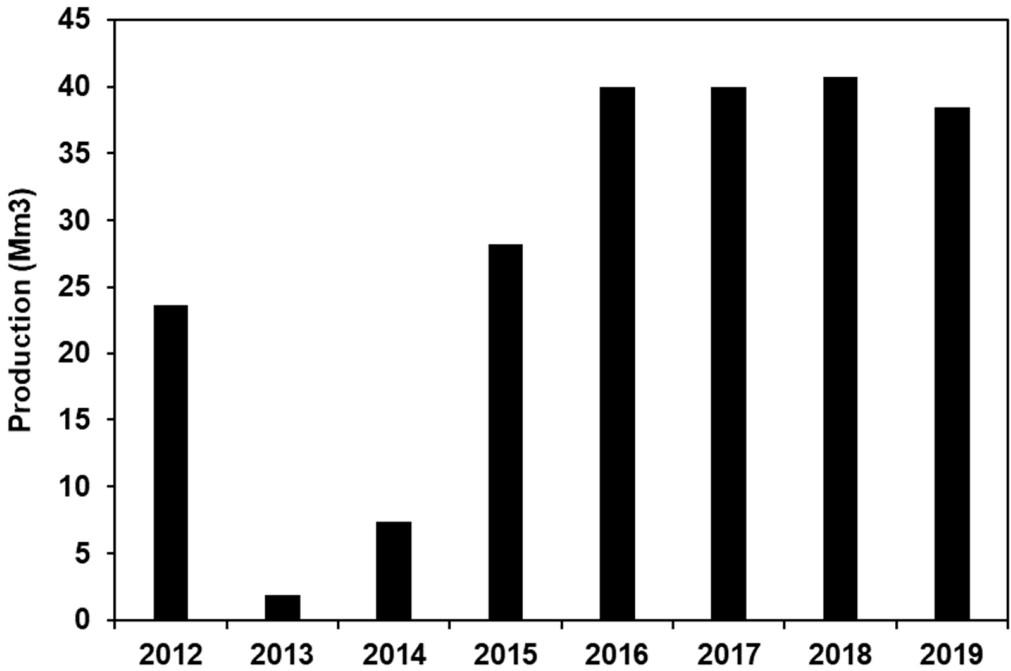

**Figure 3.** Annual production (millions m³) of freshwater by the San Pedro del Pinatar SWRO I and II plants between 2012 and 2019.

*3.2. Energy Consumption by the Pumping System*

Figure 4 presents the operating time of the pumps from 2012 to 2019. The maximum operating time was observed in 2016 and 2017.

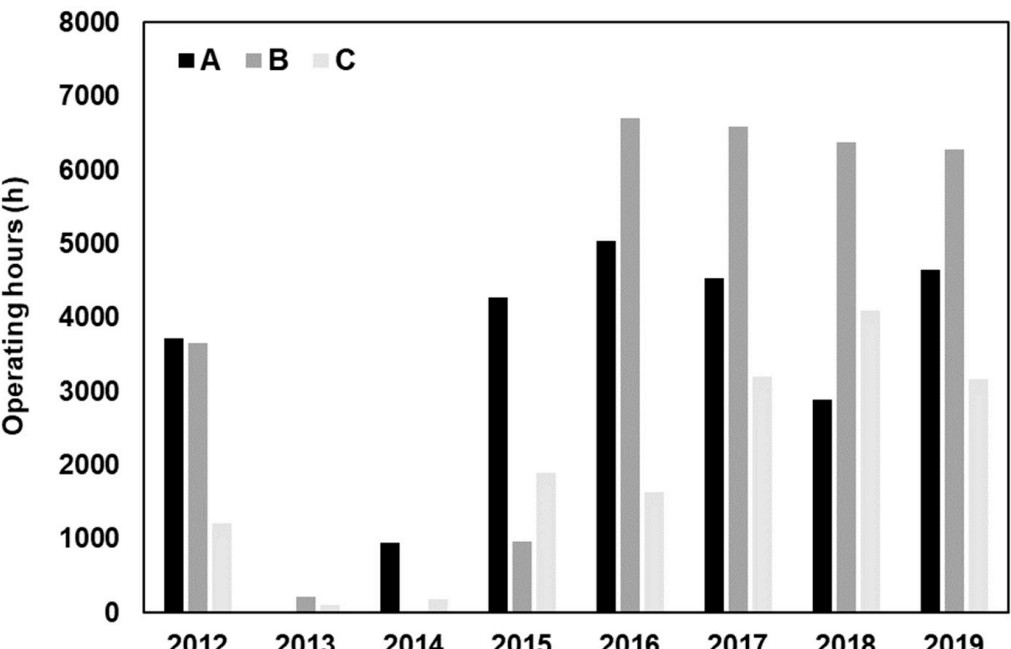

**Figure 4.** Operating hours of the pumps used to evacuate the brine discharge between 2012 and 2019. Pumps A and B have a frequency variator and pump C has a starter.

Table 3 shows the monthly energy consumption of the three pumps. High energy consumption was recorded between 2017 and 2019, with a maximum consumption of 1.24 GWh in 2017. The year with lowest energy consumption was 2013 with 0.19 GWh. The months with the highest energy consumption were August and September while the minimum consumption was in the winter months (January–March).

**Table 3.** Energy consumption (MWh) of the brine disposal pumps of the SWRO of San Pedro del Pinatar II.

| Month | 2012 | 2013 | 2014 | 2015 | 2016 | 2017 | 2018 | 2019 | Average |
|---|---|---|---|---|---|---|---|---|---|
| January | 9.0 | 1.0 | 2.6 | 7.3 | 91.0 | 11.3 | 115.7 | 100.9 | 338.6 |
| February | 6.8 | 1.7 | 0.8 | 11.1 | 80.9 | 7.4 | 50.2 | 56.5 | 215.2 |
| March | 13.0 | 4.1 | 1.6 | 4.5 | 84.7 | 104.1 | 54.3 | 68.5 | 334.9 |
| April | 12.1 | 0.9 | 0.0 | 10.8 | 120.1 | 108.3 | 149.4 | 100.5 | 502.1 |
| May | 63.5 | 0.6 | 1.9 | 24.3 | 111.6 | 151.3 | 172.5 | 36.6 | 562.4 |
| June | 72.8 | 1.5 | 4.3 | 24.8 | 113.7 | 128.3 | 79.9 | 113.8 | 538.9 |
| July | 63.3 | 3.9 | 1.9 | 20.7 | 91.2 | 99.5 | 115.2 | 119.5 | 515.1 |
| August | 48.1 | 1.9 | 1.3 | 137.0 | 123.1 | 150.7 | 108.9 | 139.5 | 710.5 |
| September | 95.6 | 0.0 | 0.8 | 66.3 | 109.0 | 123.6 | 140.2 | 136.4 | 671.8 |
| October | 56.3 | 0.1 | 3.1 | 93.7 | 85.6 | 104.0 | 87.9 | 125.9 | 556.7 |
| November | 12.2 | 0.0 | 21.6 | 47.0 | 71.6 | 139.3 | 68.3 | 117.5 | 477.6 |
| December | 1.1 | 3.8 | 36.4 | 123.5 | 11.3 | 111.3 | 44.5 | 103.8 | 435.5 |
| Total | 453.7 | 19.4 | 76.1 | 571.1 | 1093.7 | 1239.0 | 1187.0 | 1219.3 | 5859.3 |

The total energy consumption by the SWRO plants of San Pedro del Pinatar I and II is shown in Table 4. Generally high energy consumption was observed between 2016 and 2019, with a maximum consumption of 222.36 GWh in 2017. The highest energy consumption was between July and October. On the other hand, 2013 was the year with the lowest energy consumption at 9.91 GWh.

**Table 4.** Energy consumption (GWh) of the SWRO of San Pedro del Pinatar I and II.

| Month | 2012 | 2013 | 2014 | 2015 | 2016 | 2017 | 2018 | 2019 |
|---|---|---|---|---|---|---|---|---|
| January | 4.33 | 1.67 | 0.80 | 7.67 | 16.15 | 6.99 | 22.03 | 15.33 |
| February | 4.42 | 1.67 | 0.73 | 6.33 | 15.13 | 3.86 | 10.15 | 10.92 |
| March | 6.93 | 1.38 | 0.66 | 7.57 | 14.85 | 16.56 | 22.74 | 13.28 |
| April | 6.86 | 0.57 | 0.76 | 7.91 | 17.82 | 16.91 | 21.92 | 13.91 |
| May | 14.99 | 0.53 | 1.30 | 8.96 | 18.90 | 20.50 | 20.51 | 11.24 |
| June | 10.93 | 0.54 | 1.23 | 8.95 | 22.01 | 21.21 | 22.35 | 20.32 |
| July | 19.69 | 0.67 | 4.49 | 8.34 | 23.28 | 21.16 | 20.19 | 23.13 |
| August | 11.26 | 0.55 | 3.57 | 20.63 | 23.54 | 24.35 | 19.41 | 24.53 |
| September | 16.85 | 0.47 | 4.96 | 12.46 | 22.22 | 21.20 | 21.88 | 21.08 |
| October | 15.11 | 0.81 | 5.38 | 15.43 | 19.81 | 20.49 | 15.55 | 22.91 |
| November | 2.05 | 0.51 | 5.20 | 10.93 | 14.34 | 21.59 | 12.45 | 21.90 |
| December | 1.42 | 0.55 | 7.94 | 11.24 | 8.55 | 22.58 | 13.17 | 16.37 |
| Total | 114.84 | 9.91 | 37.0 | 126.39 | 216.61 | 217.40 | 222.36 | 214.92 |

*3.3. Economic Cost*

Table 5 shows the economic cost associated with the pumping system. The economic cost is proportional to the percentage of energy consumed by the pumps. The highest economic cost was observed from July to November, with a monthly average of 5150 €. From 2012 to 2019, the total cost of energy consumption by the pumps was 394,682 €. The cost has increased since 2013, reaching a maximum of 89,390 € in 2019.

**Table 5.** Energy cost (€) for brine discharge disposal of the SWRO of San Pedro del Pinatar I and II.

| Month | 2012 | 2013 | 2014 | 2015 | 2016 | 2017 | 2018 | 2019 | Average (€) |
|---|---|---|---|---|---|---|---|---|---|
| January | 573.8 | 80.3 | 182.7 | 480.0 | 7116.9 | 928.7 | 8648.1 | 7714.7 | 3215.7 |
| February | 430.6 | 136.4 | 49.2 | 726.0 | 6577.9 | 540.1 | 3614.6 | 4625.4 | 2087.5 |
| March | 780.9 | 281.3 | 132.1 | 287.5 | 3707.2 | 7276.1 | 2926.6 | 4671.8 | 2507.9 |
| April | 700.0 | 60.2 | - | 617.2 | 4466.0 | 7449.1 | 8204.8 | 6462.3 | 3994.2 |
| May | 4013.7 | 39.0 | 121.3 | 1505.4 | 4369.5 | 10,607.2 | 11,292.6 | 2353.9 | 4287.8 |
| June | 4975.6 | 99.5 | 284.7 | 1583.0 | 6233.1 | 9661.0 | 5819.1 | 8241.7 | 4612.2 |
| July | 6076.1 | 269.2 | 150.3 | 1338.1 | 5320.7 | 7531.6 | 8897.2 | 9247.1 | 4853.8 |
| August | 2897.5 | 113.0 | 78.9 | 8216.1 | 6349.5 | 8705.0 | 6717.0 | 8548.9 | 5203.2 |
| September | 6707.8 | - | 53.5 | 5015.0 | 6161.6 | 7596.0 | 9640.9 | 10,333.2 | 6501.1 |
| October | 3754.3 | 8.0 | 214.1 | 6747.8 | 5624.2 | 7199.3 | 5682.8 | 9200.9 | 4803.9 |
| November | 852.1 | - | 1503.5 | 3465.4 | 5007.7 | 10,101.8 | 4707.6 | 8861.9 | 4928.6 |
| December | 86.6 | 249.3 | 2838.4 | 9131.3 | 843.0 | 8472.3 | 3334.9 | 9128.6 | 4260.6 |
| Total (€) | 31,849.0 | 1336.2 | 5608.7 | 39,112.8 | 61,777.3 | 86,068.2 | 79,486.2 | 89,390.4 | |

The energy costs for the SWRO plants at San Pedro del Pinatar I and II is shown in Table 6. The highest cost of 12.4 M€ was incurred in 2017, while the lowest cost of 1.11 M€ was observed in 2013.

**Table 6.** Cost of the energy consumed by the SWRO plants of San Pedro del Pinatar (Thousand €).

| Month | 2012 | 2013 | 2014 | 2015 | 2016 | 2017 | 2018 | 2019 | Average |
|---|---|---|---|---|---|---|---|---|---|
| January | 673.92 | 86.76 | 47.20 | 541.64 | 899.42 | 689.52 | 443.33 | 679.90 | 673.92 |
| February | 470.07 | 84.60 | 44.21 | 826.48 | 406.30 | 1033.88 | 424.35 | 171.82 | 470.07 |
| March | 2056.27 | 77.72 | 52.94 | 735.28 | 503.38 | 902.75 | 865.70 | 682.87 | 2056.27 |
| April | 1376.52 | 52.46 | 67.65 | 559.47 | 510.79 | 1238.42 | 840.64 | 717.85 | 1376.52 |
| May | 629.70 | 74.23 | 116.49 | 909.56 | 538.13 | 1136.66 | 977.07 | 675.73 | 629.70 |
| June | 199.86 | 61.70 | 126.00 | 760.63 | 862.21 | 1122.44 | 1139.47 | 1059.19 | 199.86 |
| July | 1304.45 | 57.44 | 233.72 | 551.43 | 957.18 | 1149.38 | 1131.62 | 1256.15 | 1304.45 |
| August | 690.11 | 80.96 | 320.19 | 791.71 | 865.34 | 957.16 | 933.80 | 1029.96 | 690.11 |
| September | 873.12 | 50.37 | 104.87 | 437.48 | 896.51 | 894.99 | 1082.46 | 1082.69 | 873.12 |
| October | 800.11 | 71.97 | 214.80 | 868.97 | 963.54 | 1030.77 | 818.50 | 1118.93 | 800.11 |
| November | 100.88 | 226.48 | 389.75 | 731.98 | 833.97 | 1087.11 | 737.90 | 1139.06 | 100.88 |
| December | 72.41 | 188.56 | 718.89 | 683.99 | 621.50 | 1165.35 | 783.30 | 1036.13 | 72.41 |
| Total | 9247.42 | 1113.26 | 2436.71 | 8398.62 | 8858.26 | 12,408.43 | 10,178.14 | 10,650.29 | 9247.42 |

The consumption of the discharge pumping system varies from 10 to 30 kWh for each 1000 m$^3$ of water produced by the desalination plants. The energy consumption of the discharge pumping is directly related to the production of desalinated water; thus, higher brine discharge production from the San Pedro del Pinatar SWRO plants implies higher pumping energy (Figure 5). Figure 6 shows a comparison of the energy consumption of the brine disposal pumps and both desalination plants and computes the percentage of the economic cost of this consumption. The economic cost was highest in 2019, associated with a percentual energy consumption of 0.843%, and the lowest value was found for 2014 with a percentage of 0.229%. The energy consumption was highest in 2019 with a percentage of 0.962%, and the lowest value was found for 2013 with a 0.17%.

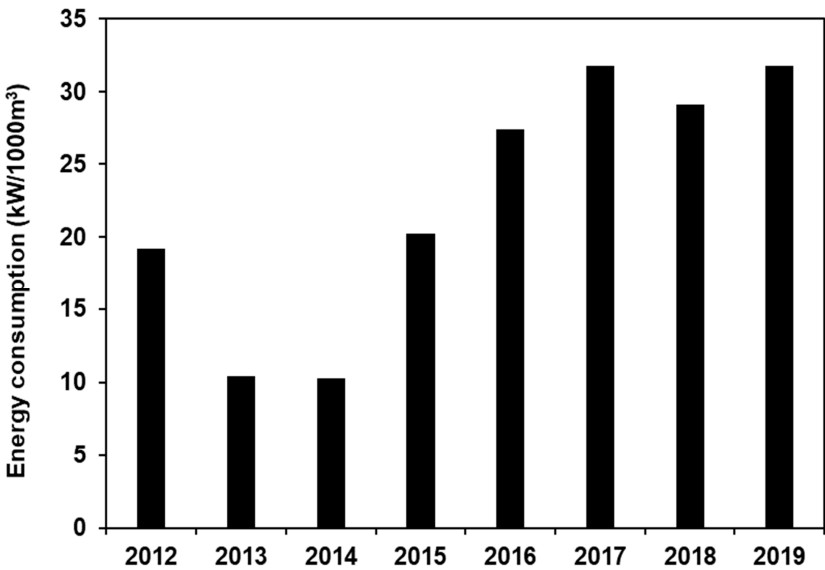

**Figure 5.** Energy consumption of pumping system per 1000 m³ of water produced between 2012 and 2019.

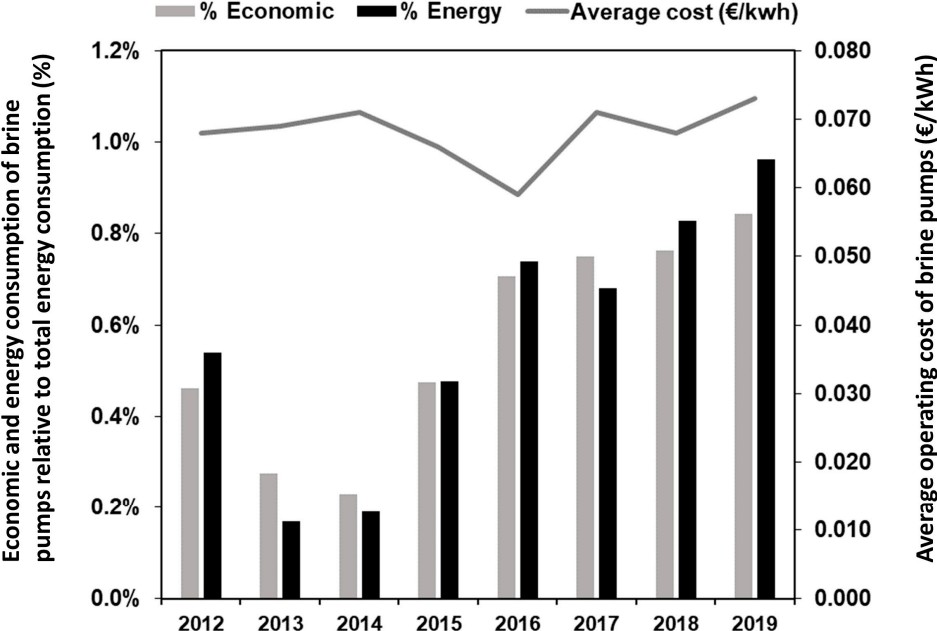

**Figure 6.** Annual percentage of economic and energy consumption of the brine disposal pumps related to the total energy consumption of the SWRO (second *y*-axis) between 2012 and 2019. Annual average cost of brine disposal pumps (*y*-axis).

## 4. Discussion

Our work presents the first systematic evaluation of the energetic and economic cost of discharging brine safely into the marine environment by incorporating a diffuser piece at the end of a submerged outfall. The novelty of this study is to evaluate for the first time the extra energy and economic cost associated with the installation of a brine pumping system related with the installation of a diffuser piece on the submerged outfall. The arrangements at San Pedro del Pinatar I and II SWRO plants are used as a case study. The use of a diffuser promoted the recovery of the abundance and diversity of the benthic fauna affected by the brine discharge [8] at a low economic and energy cost. The results indicate that there

are cost-effective means to meet environmental protection requirements and minimize the environmental impact associated with the discharged brine.

The diffuser piece installed in the brine discharge of San Pedro del Pinatar I and II SWRO plants requires higher pressure and higher velocity pumping of the effluent to maximize the brine discharge dilution in the marine environment [14]. The results showed that, from 2012 to 2019, the operation of these pumps consumed only 0.229% to 0.843% of the total energy required for the operation of the San Pedro SWRO plants. In addition, this energy consumption amounted to between 10 and 30 kWh per thousand cubic meters produced.

The costs incurred for discharging the brine ranged from 1336 € in 2013 to 89,390 € in 2019. Energy consumption and economic cost are directly related to the production of desalinated water; thus, higher output from the San Pedro del Pinatar SWRO plants implied more brine discharge and higher pumping energy [1,12]. Since 2015, the freshwater production has increased to approximately 20 Mm$^3$, representing an increase of approximately 42% in the maximum freshwater production capacity of both plants. The annual variability observed in the fresh water production has been explained with the availability of water from different sources and prioritizing the cheapest water resources available [1,21]. Therefore, there was a significant reduction in desalinated water production in 2013 and 2014 due to the abundance of water from other resources during these years [21].

On the other hand, the operation of the pumps made up 0.23% and 0.84% of the total energy consumption of the plant from 2012 to 2020, with an average value of only 0.563% of the total energy cost of the San Pedro SWRO plants. This indicates that the use of diffusers is an effective measure that can be adopted by SWRO plants for complying with environmental regulations in a cost- and energy-effective manner.

The results can be compared with other methods described in the literature. The desalination plants at Alicante use a seawater bypassing system to comply with the environmental regulations and to reduce the influence area of brine. An average additional cost of 1.7% was incurred [16]. Additionally, these plants use an irrigation program to maintain the groundwater level in the saltmarsh and compensate for the effect of the intake system on the saltmarsh, but with lower energy consumption than the dilution system [22]. The results obtained in this study show that the use of a diffuser piece in the outfall represents an average energy consumption of 0.57%, which is considerably lower than the seawater bypassing method used in Alicante SWRO plants with similar desalinated water production capacity. Moreover in the Javea Desalination Plant with a constant dilution ratio of 4 parts of water to 1 part of brine, the seawater bypassing system also has higher energy consumption [13].

The installation of the brine discharge pumping system in the San Pedro del Pinatar SWRO plants, in combination with the use and configuration of the diffuser piece on the submerged outfall, ensures effective dilution of the discharged brine in the surrounding environment and drastically reduces the affected area [14,20,23,24]. The installation of diffuser pieces is deemed one of the best methods globally to minimize the environmental impact of brine discharge and maximize the dilution of the brine discharge influence area [15,25,26].

Finally, this study demonstrates that using a diffuser piece system to meet environmental requirements entails a very low economic cost for SWRO plants. Further, using diffusers entails a lower economic cost than using other methods, such as seawater bypassing systems [16].

## 5. Conclusions

The results obtained in this study show that the installation of diffuser devices in submerged outfalls is a means of ensuring sustainable desalination development with low economic and energetic costs [13,27,28]. Energy consumption of the brine discharge pumping system in the period under study oscillated between 0.229% and 0.843% of the total energy required for the operation of the San Pedro SWRO plants, which represents

an average value of energy costs of 0.56% of the total energy cost for both plants. For future research, this study could be extended to compare energy costs in other SWRO plants, using the same disposal method to meet environmental requirements, but with different characteristics. The findings obtained in this study could be extrapolated to other SWRO plants in other world regions, in order to improve brine discharge management in currently operating plants or future SWRO projects. As the demand for desalination operations increases, economically and scientifically viable technologies, such as diffusers in SWRO plants to mitigate environmental impact, are necessary to advance towards global sustainable desalination development.

**Author Contributions:** Conceptualization, R.N. and J.L.S.L.; methodology, R.N. and I.S.; investigation, R.N. and I.S.; writing—original draft preparation, I.S. and R.N.; writing—review and editing, I.S. and J.L.S.L.; supervision, J.L.S.L. All authors have read and agreed to the published version of the manuscript.

**Funding:** This research received no external funding.

**Data Availability Statement:** Data have been included in the manuscript.

**Conflicts of Interest:** The authors declare that the research was conducted in the absence of any commercial or financial relationships that could be construed as a potential conflict of interest.

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
