# Peer review of "Assessment of Energy Consumption of Brine Discharge from SWRO Plants"

_water, doi:10.3390/w15040786_

Round 1
Reviewer 1 Report
Improve substantially paper research, including innovation
Author Response
We have improved the paper according reviewers’ recommendations and answering every comment.
Reviewer 2 Report
Please see attached pdf file.

Author Response
Dear Editor and reviewer,
Please find attached the response to reviewer's document and the revised copy of the manuscript “Assessment of energy consumption of brine discharge from SWRO plant” by R. Navarro et al based on the reviewers’ recommendations and the editor evaluation. All reviewers’ recommendations have been considered for better definition of our aims and improvement of the manuscript.
Additionally we have modified the order of authors and the corresponding author of the manuscript. Here you can find detailed answer to every reviewer comment.

Reviewer 3 Report
The article is a scientific approach to a case study concerning desalination stations in Spain, which were built near meadows protected at the national and European level. The aim of the work was to assess the economic costs of energy used for the process of brine discharge, necessary to meet environmental requirements. Was there anything new about this? Economic evaluation of the activity is a fairly common activity in every activity? A good discussion of the results deserves to be emphasized.
The article is not bad, but it needs some additions.
1. Please complete the purpose and novelty of the work. It must result from the analysis of the literature and emphasize the authors' contribution
2. in the conclusions, it is worth indicating further research plans resulting from the conclusions of the research
3. can the results be generalized? Are there any universal conclusions from the research? whether it is only the result of a case study - it is also worth writing down in the conclusions
Author Response
Dear Editor and reviewer,
Please find attached the response to reviewer's document and the revised copy of the manuscript “Assessment of energy consumption of brine discharge from SWRO plant” by R. Navarro et al based on the reviewers’ recommendations and the editor evaluation. All reviewers’ recommendations have been considered for better definition of our aims and improvement of the manuscript.
Additionally we have modified the order of authors and the corresponding author of the manuscript. Here you can find detailed answer to every reviewer comment.
The manuscript has been carefully reviewed by an experienced editor whose first language is English and who specializes in editing papers written by scientists whose native language is not English. We have included a certificate of editing.

Round 2
Reviewer 1 Report
Must be improved the technical paper in order to get quality of research contribution. Practical case is interesting, but enough even there is not any comparation with other similar solutions.
Author Response
We have tried to improve the manuscript in order to highlight the research done. This is, to our knowledge, the first time that the energy cost of the brine discharge though a pipeline has been estimated and the implications for desalination plant operation are discussed. It has been compared with other similar solutions to increase the dilution of the brine discharge as seawater by-passing including new references.
Yours sincerely

Round 3
Reviewer 1 Report
New contributions made are insufficient.
Unfortunately doesn´t provide highlight research content